# Early structural and functional plasticity alterations in a susceptibility period of DYT1 dystonia mouse striatum

**Marta Maltese[1,2], Jennifer Stanic[3], Annalisa Tassone[1,2], Giuseppe Sciamanna[1,2], Giulia Ponterio[1,2], Valentina Vanni[1,2], Giuseppina Martella[1,2], Paola Imbriani[1,2], Paola Bonsi[2], Nicola Biagio Mercuri[1,2], Fabrizio Gardoni[3†], Antonio Pisani[1,2†]***

[1]Department of Systems Medicine, University of Rome Tor Vergata, Rome, Italy; [2]IRCCS Fondazione Santa Lucia, Rome, Italy; [3]Department of Pharmacology, University of Milan, Milan, Italy

**Abstract** The onset of abnormal movements in DYT1 dystonia is between childhood and adolescence, although it is unclear why clinical manifestations appear during this developmental period. Plasticity at corticostriatal synapses is critically involved in motor memory. In the $Tor1a^{+/\Delta gag}$ DYT1 dystonia mouse model, long-term potentiation (LTP) appeared prematurely in a critical developmental window in striatal spiny neurons (SPNs), while long-term depression (LTD) was never recorded. Analysis of dendritic spines showed an increase of both spine width and mature mushroom spines in $Tor1a^{+/\Delta gag}$ neurons, paralleled by an enhanced AMPA receptor (AMPAR) accumulation. BDNF regulates AMPAR expression during development. Accordingly, both proBDNF and BDNF levels were significantly higher in $Tor1a^{+/\Delta gag}$ mice. Consistently, antagonism of BDNF rescued synaptic plasticity deficits and AMPA currents. Our findings demonstrate that early loss of functional and structural synaptic homeostasis represents a unique endophenotypic trait during striatal maturation, promoting the appearance of clinical manifestations in mutation carriers.

DOI: https://doi.org/10.7554/eLife.33331.001

***For correspondence:**
pisani@uniroma2.it

[†]These authors contributed equally to this work

**Competing interests:** The authors declare that no competing interests exist.

## Introduction

Early-onset generalized torsion dystonia (DYT1) is an autosomal dominant movement disorder, commonly caused by a GAG base-pair deletion in the TOR1A gene coding for torsinA protein, without gross brain structural defects or other detectable neuropathology (*Ozelius et al., 1997*; *Ledoux et al., 2013*). Intriguingly, only 30–40% of DYT1 mutation carriers develop dystonia, typically in childhood-early adolescence (*Bressman et al., 2000*). However, what triggers the clinical onset of symptoms is currently unknown, although the presence of a critical developmental period of susceptibility is highly probable, since mutation carriers that do not develop symptoms in that time-window remain unaffected for their entire life (*Pappas et al., 2014*).

Plasticity changes include functional and structural synaptic specialization, leading to experience-dependent acquisition of motor skills. However, genetic or acquired alterations may lead to maladaptive plasticity changes. Accordingly, human studies indicate neural processing and synaptic plasticity alterations as major determinants in dystonia pathophysiology (*Quartarone and Hallett, 2013*). A significantly enhanced responsiveness to plasticity protocols has been reported in dystonic patients (*Edwards et al., 2006*; *Weise et al., 2006*; *Quartarone et al., 2009*). Moreover, patterns of impaired motor learning have been described even in clinically unaffected DYT1 mutation carriers (*Ghilardi et al., 2003*), further supporting the notion that aberrant plasticity represents a unique endophenotype in dystonia.

Of note, an impairment of striatal plasticity has been demonstrated in a number of different DYT1 models, including transgenic mice and rats overexpressing mutant torsinA (*Martella et al., 2009*; *Grundmann et al., 2012*), knock-in mice heterozygous for Δgag-torsinA (*Dang et al., 2012*; *Martella et al., 2014*; *Rittiner et al., 2016*), revealing an impressive similarity with studies of synaptic plasticity in human dystonia. Collectively, these observations support the hypothesis that DYT1 dystonia is a complex neurodevelopmental disorder of abnormal neurochemistry, wiring, and physiology (*Goodchild et al., 2013*; *Pappas et al., 2014*).

However, these alterations were observed in adult rodents, and to date, a relationship between age and corticostriatal plasticity in dystonia is still lacking. Furthermore, the question as to whether functional and structural plasticity abnormalities occur early in life or later as adaptive changes remains unknown. We report structural and functional abnormalities occurring in a defined postnatal time-window in $Tor1a^{+/\Delta gag}$ mice, indicative of a 'premature' and abnormal functional and structural plasticity, which is paralleled by a time-dependent increase in both BDNF levels and AMPAR-mediated currents.

Our findings reveal molecular, functional and structural changes in DYT1 striatal spiny projection neurons (SPNs), emphasizing the link between abnormal plasticity and dystonia. Understanding the key stages at which synaptic circuits are affected could suggest new routes to prevent or treat the disorder.

## Results

The critical period for symptom onset in DYT1 dystonia matches a time-window of postnatal life when motor memories are shaped by activity-dependent changes in the striatum. Thus, in order to characterize plasticity changes in the early adolescence, $Tor1a^{+/\Delta gag}$ mice were recorded from postnatal day P15 to P35, in good agreement with the approximate life phase equivalencies between humans and mice, predicting that ~4 weeks of mouse age correspond to ~14 years in humans (*Flurkey et al., 2007*).

### Electrophysiological characterization of SPNs

Properties of adult $Tor1a^{+/\Delta gag}$ SPNs have been extensively characterized (*Maltese et al., 2014*; *Martella et al., 2014*). Here, we focused on intrinsic and synaptic properties of juvenile $Tor1a^{+/\Delta gag}$ neurons. SPNs recorded at P26 from both $Tor1a^{+/+}$ and $Tor1a^{+/\Delta gag}$ mice did not display firing activity at rest and exhibited no significant differences in their intrinsic membrane properties (data not shown). Depolarizing and hyperpolarizing current steps caused tonic action potential discharge and strong inward membrane rectification (*Figure 1A*). Short ISI (25–50 ms) of paired synaptic stimulation induced PPF in both genotypes (*Figure 1B*; p<0.05). At longer ISI (100–1000 ms), PPF was not observed in juvenile $Tor1a^{+/+}$ and $Tor1a^{+/\Delta gag}$ mice (*Figure 1B*; p>0.05). To explore potential differences in neurotransmitter release, we recorded spontaneous glutamate- and GABA-mediated currents in P26 SPNs from both $Tor1a^{+/+}$ and $Tor1a^{+/\Delta gag}$ mice. Glutamatergic sEPSCs did not differ between genotypes (*Figure 1C*; p>0.05). However, we found a significant increase in the amplitude, but not in the frequency, of mEPSCs recorded from $Tor1a^{+/\Delta gag}$ mice compared to wild types (*Figure 1D*; p<0.05). Conversely, GABAergic sIPSCs were unchanged in $Tor1a^{+/\Delta gag}$ with respect to $Tor1a^{+/+}$ littermates (*Figure 1E*; p>0.05). Also, mIPSCs were similar in both genotypes (*Figure 1F*; p>0.05).

### Premature expression of corticostriatal synaptic plasticity

We previously demonstrated a marked impairment of bidirectional synaptic plasticity in adult (P60-P75) $Tor1a^{+/\Delta gag}$ striatum (*Martella et al., 2014*). However, it remains unclear whether these patterns of abnormal plasticity are core pathologic features in an early developmental period, or occur later as maladaptive changes. Thus, we performed a detailed characterization of LTD and LTP from P15 to P35 in $Tor1a^{+/+}$ and $Tor1a^{+/\Delta gag}$ mice. In $Tor1a^{+/+}$ SPNs, HFS failed to induce LTD from P15 to P27 (*Figure 2A*; p>0.05). Conversely, the HFS protocol elicited a robust LTD from P28 to P35 (*Figure 2A*; 59.63 ± 2.63% of control; p<0.05). Surprisingly, in slices from $Tor1a^{+/\Delta gag}$ mice, HFS stimulation failed to cause a synaptic depression, independently from the postnatal day of recording (*Figure 2B*; p>0.05).

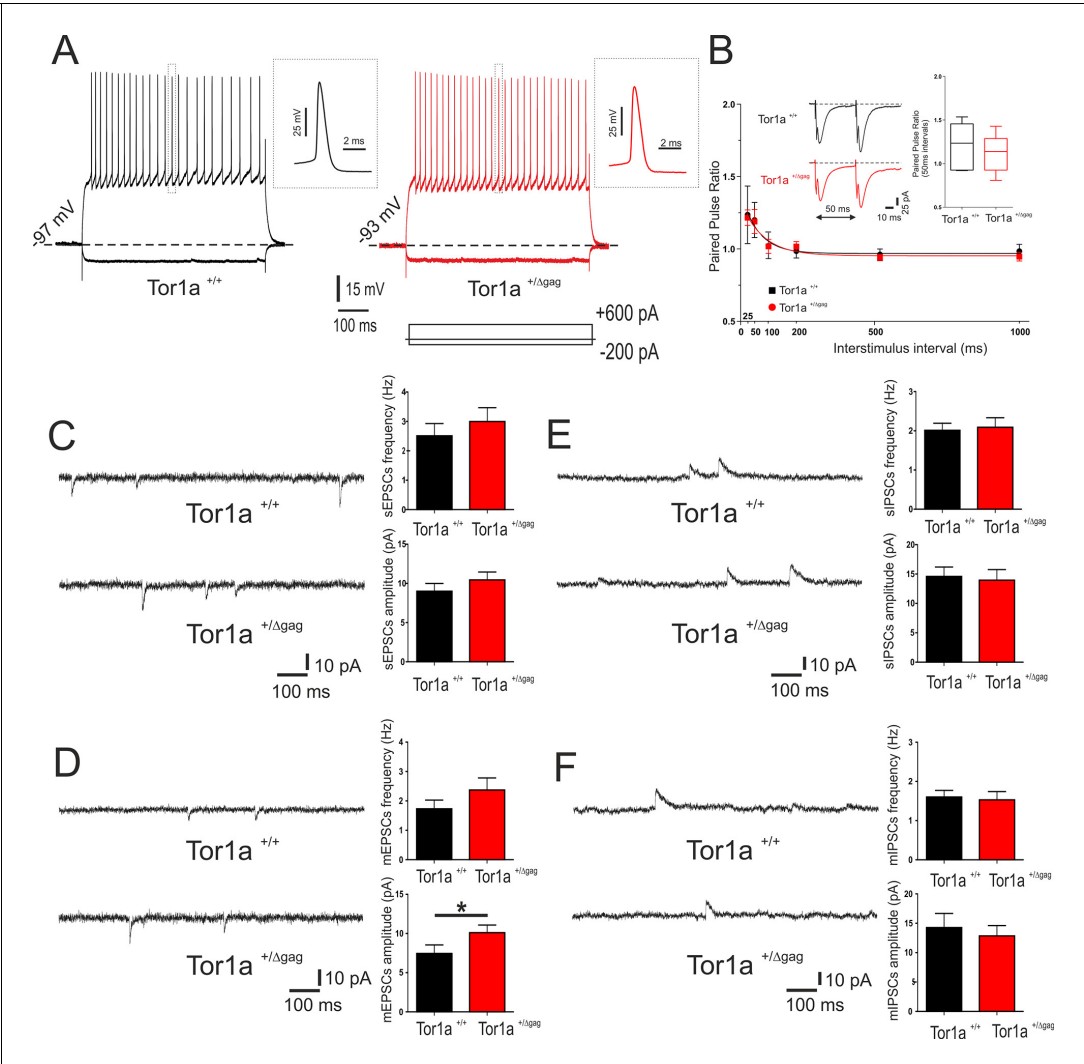

**Figure 1.** Electrophysiological and synaptic properties of striatal SPNs. (**A**) Superimposed traces showing voltage responses to both depolarizing (+600 pA) and hyperpolarizing (−200 pA) current steps in SPNs recorded from P26 *Tor1a*$^{+/+}$ (black) and *Tor1a*$^{+/\Delta gag}$ (red) mice. The insets display single action potentials (amplitude: *Tor1a*$^{+/+}$69.62 ± 1.14 mV, N = 11, n = 11; *Tor1a*$^{+/\Delta gag}$66.65 ± 1.68 mV, N = 8, n = 11; Student's t test p>0.05). (**B**) Summary plot of paired-pulse ratio values showing similar facilitation in both genotypes. Each data point represents mean ± SEM. P26 *Tor1a*$^{+/+}$ mice N = 3, 25 ms: 1.24 ± 0.20, n = 5; 50 ms: 1.20 ± 0.12, n = 5, Student's t test p<0.05; P26 *Tor1a*$^{+/\Delta gag}$ mice N = 3, 25 ms: 1.22 ± 0.05, n = 5; 50 ms: 1.19 ± 0.08, n = 5; Student's t test p<0.05. Insets represent sample traces showing facilitation at ISI = 50 ms in both genotypes. (**C**) Representative sEPSCs recordings in PTX from SPNs of P26 *Tor1a*$^{+/+}$ and *Tor1a*$^{+/\Delta gag}$ mice. HP: −70 mV. The summary plots show no significant difference between genotypes in sEPSCs frequency and amplitude (Student's t test p>0.05). (**D**) Representative whole-cell recordings in PTX plus TTX of mEPSC from P26 *Tor1a*$^{+/+}$ and *Tor1a*$^{+/\Delta gag}$ SPNs. HP: −70 mV. Plots show a significant difference in the amplitude of mEPSCs recorded from *Tor1a*$^{+/\Delta gag}$ mice compared to wild-types (*Tor1a*$^{+/+}$, 7.45 ± 1.09, N = 9, n = 9; *Tor1a*$^{+/\Delta gag}$, 10.11 ± 0.97, N = 8, n = 9; Student's t test *p<0.05). (**E**) Representative recordings in MK-801 and CNQX of sIPSCs from P26 *Tor1a*$^{+/+}$ and *Tor1a*$^{+/\Delta gag}$ SPNs. HP:+10 mV. The summary plots show no significant difference in sIPSC frequency and amplitude (Student's t test p>0.05). (**F**) Representative traces of mIPSCs recorded in MK-801, CNQX and TTX. HP:+10 mV. The summary plots show no difference in frequency and amplitude between genotypes (Student's t test p>0.05). Data are presented as mean ± SEM.

DOI: https://doi.org/10.7554/eLife.33331.002

The following source data is available for figure 1:

**Source data 1.** Electrophysiological and synaptic properties of striatal SPNs.

DOI: https://doi.org/10.7554/eLife.33331.003

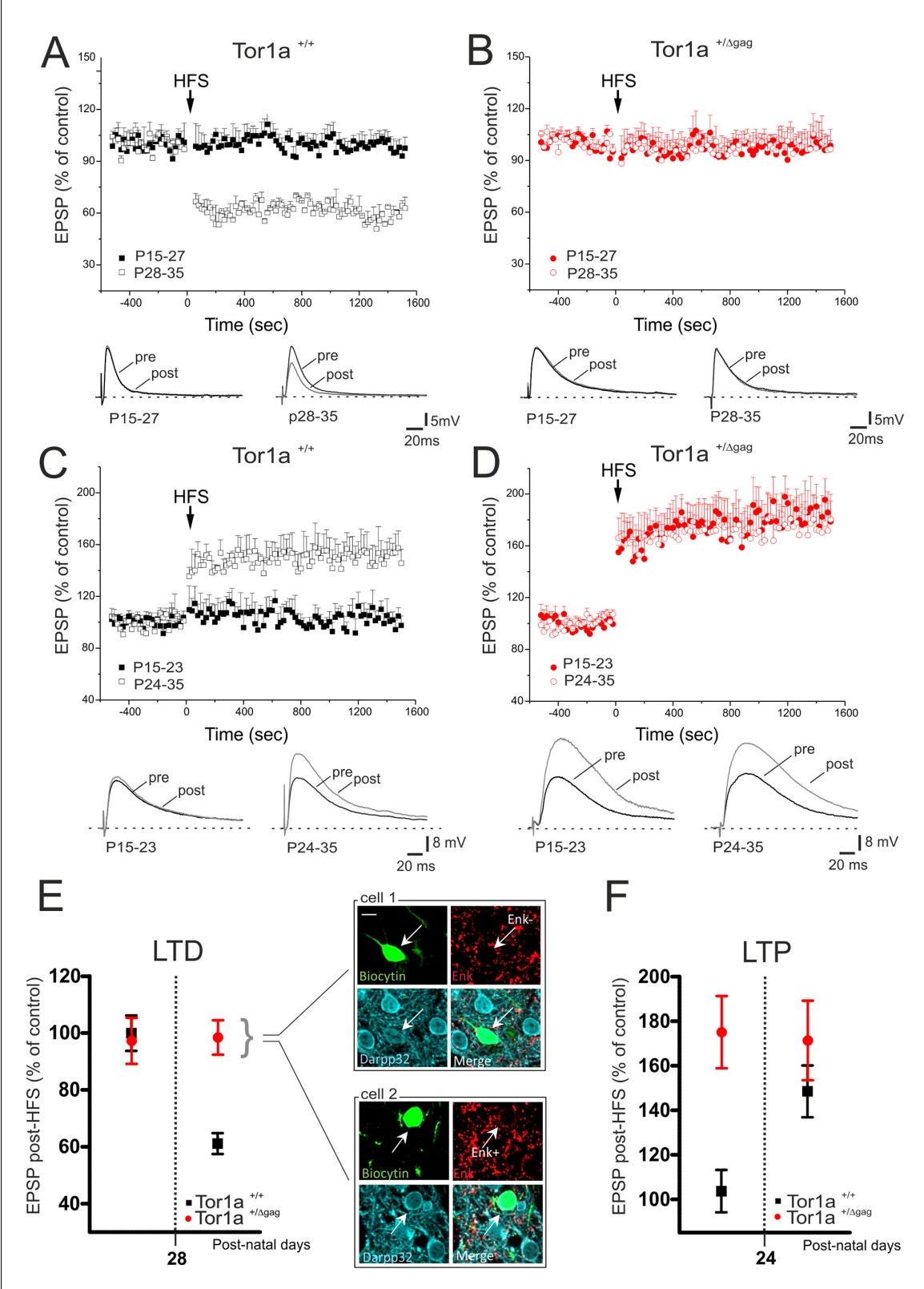

**Figure 2.** Altered developmental profile of corticostriatal long-term synaptic plasticity expression in *Tor1a*$^{+/\Delta gag}$ mice. (**A**) (*Top*) Developmental time-course of LTD expression in *Tor1a*$^{+/+}$ mice. HFS protocol (arrow) induces LTD in SPNs recorded from *Tor1a*$^{+/+}$ mice after P28 (59.63 ± 2.63% of control; N = 8, n = 8; paired Student's t test p<0.05), but not from P15 to P27 (99.46 ± 4.65, N = 9, n = 10; paired Student's t test p>0.05). (*Bottom*) Representative EPSP traces recorded before (pre) and 20 min after (post) HFS protocol delivery. (**B**) (*Top*) In *Tor1a*$^{+/\Delta gag}$ mice, HFS protocol fails to

*Figure 2 continued on next page*

Figure 2 continued

induce any LTD, irrespective of the postnatal age (P15-27, 96.85 ± 11.35% of control; N = 8, n = 12; P28-35, 100.29 ± 4.16% of control, N = 8, n = 12; paired Student's t test p>0.05). (Bottom) Representative traces of EPSPs recorded pre- and post-HFS. (C) (Top) Time-course of corticostriatal LTP expression during postnatal development in $Tor1a^{+/+}$ mice. HFS of corticostriatal afferents (arrow) induces LTP expression in $Tor1a^{+/+}$ mice after P24 (148.80 ± 15.39% of control; N = 6, n = 10; paired Student's t test p<0.05), but not at P15-23 (104.68 ± 8.99% of control; N = 6, n = 10; paired Student's t test p>0.05). (Bottom) Sample EPSPs recorded pre- and post-HFS protocol in $Tor1a^{+/+}$ mice. (D) (Top) SPNs recorded from $Tor1a^{+/\Delta gag}$ mice exhibit a premature LTP (P15-23, 174.68 ± 22.59% of control; N = 6, n = 10; P24-35, 172.35 ± 11.06% of control; N = 9, n = 10; paired Student's t test p<0.05). (Bottom) EPSP traces recorded pre- and post-LTP induction. (E) Mean plot comparing LTD expression at different postnatal days in $Tor1a^{+/+}$ and $Tor1a^{+/\Delta gag}$ SPNs. (Inset) Confocal imaging of two SPNs recorded from $Tor1a^{+/\Delta gag}$ slices filled with biocytin (green) and immunolabelled for ENK (red) and DARPP-32 (cyano), marker of SPNs. Both ENK-positive and ENK-negative biocytin-labeled SPNs showed lack of LTD (scale bar: 10 μm). (F) Mean plot comparing LTP expression at different postnatal days in $Tor1a^{+/+}$ and $Tor1a^{+/\Delta gag}$ SPNs. Values are presented as mean ± SEM.

DOI: https://doi.org/10.7554/eLife.33331.004

The following source data is available for figure 2:

Source data 1. Altered developmental profile of corticostriatal long-term synaptic plasticity expression in $Tor1a^{+/\Delta gag}$ mice.
DOI: https://doi.org/10.7554/eLife.33331.005

The LTP induction protocol failed to elicit a potentiation in $Tor1a^{+/+}$ mice from P15 to P23 (**Partridge et al., 2000**) (**Figure 2C**; p>0.05), whereas a stable LTP occurred from P24 to P35 (**Figure 2C**; 148.80 ± 15.39% of control; p<0.05). Unexpectedly, in $Tor1a^{+/\Delta gag}$ SPNs LTP could be evoked as early as P15, revealing a premature onset, and showed a tendency to increase, compared to wild types (**Figure 2D**; $Tor1a^{+/\Delta gag}$ P15-23, 174.68 ± 22.59% of control; P24-35, 172.35 ± 11.06% of control; p<0.05).

The pattern of torsinA expression is common to all striatal DARPP-32-labeled neurons (**Martella et al., 2009**). To unmask potential differences between direct- and indirect-pathway SPNs, recording electrodes were filled with biocytin. Enkephalin staining revealed that neither ENK-positive nor ENK-negative SPNs exhibited LTD, ruling out a possible segregation to a specific population of SPNs (**Figure 2E**).

Collectively, these data demonstrate that LTD appeared at P28 in wild-type mice, whereas it could not be elicited during the entire postnatal period of observation in $Tor1a^{+/\Delta gag}$ mice (**Figure 2E**). Moreover, while in $Tor1a^{+/+}$ mice LTP could not be evoked before P24, in SPNs from $Tor1a^{+/\Delta gag}$ LTP appeared prematurely at P15 (**Figure 2F**).

## Increased AMPA receptor function and abundance at corticostriatal synapses during development

Changes in synaptic strength during learning and memory processes implicate an accurate regulation of AMPARs and NMDARs expression at postsynaptic membranes (**Bassani et al., 2013**; **Czöndör and Thoumine, 2013**). Thus, we performed an electrophysiological and biochemical characterization of AMPARs and NMDARs of SPNs in both $Tor1a^{+/+}$ and $Tor1a^{+/\Delta gag}$ mice.

To investigate the relative abundance of postsynaptic AMPARs and NMDARs, NMDAR/AMPAR current ratios at corticostriatal synapses were evaluated in both juvenile (P26) and adult (P60) $Tor1a^{+/+}$ and $Tor1a^{+/\Delta gag}$ SPNs (**Figure 3A–B**). We found that, at P26, the NMDAR/AMPAR ratio was significantly reduced in $Tor1a^{+/\Delta gag}$ SPNs compared to wild types (**Figure 3A**; p<0.05). Conversely, no significant differences were recorded in P60 SPNs of both genotypes (**Figure 3B**; p>0.05). A reduced NMDAR/AMPAR ratio could reflect an increase in AMPAR function or number, a decrease in NMDARs function, or even a combination of both. To detect possible differences in the composition of postsynaptic glutamate receptors in P26 SPNs, a IV relationship of AMPAR-EPSC was recorded (**Figure 3C**). $Tor1a^{+/\Delta gag}$ SPNs showed a significantly increased current at hyperpolarized voltage ranges (**Figure 3C**; 2-way ANOVA, p<0.01;HP= −70 mV). The GluA2 subunit reduces AMPAR permeability to $Ca^{2+}$. Therefore, depending on the subunit composition, AMPAR-EPSC may show a linear or an inward-rectifying IV relationship (**Cull-Candy et al., 2006**). Thus, we measured the rectification index (RI), calculated as the ratio between the AMPAR-EPSC at −70 mV and at +40 mV (**Isaac et al., 2007**). We observed no significant difference in RI between genotypes (**Figure 3C**; p>0.05), suggesting that the enhanced AMPAR current involves an increased surface expression of AMPARs, rather than an altered receptor composition. Moreover, the AMPAR-EPSC IV relationship was also recorded in the presence of the selective antagonist of GluA2-lacking AMPARs, NASPM

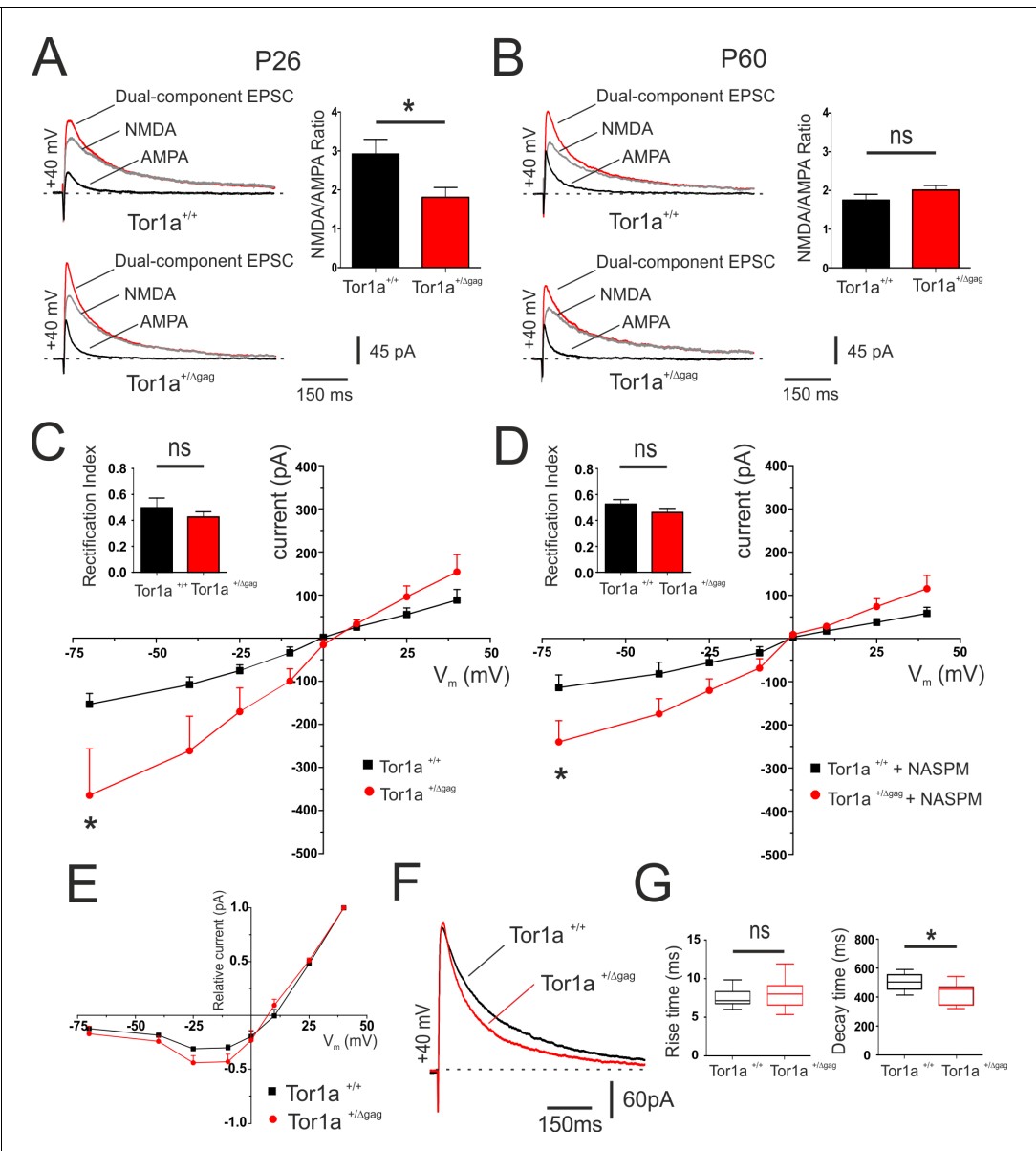

**Figure 3.** Electrophysiological characterization of AMPAR and NMDAR currents at corticostriatal synapses of SPNs in both *Tor1a*$^{+/+}$ and *Tor1a*$^{+/\Delta gag}$ mice. (A) (*Left*) Representative EPSCs traces recorded at HP=+40 mV from SPNs of juvenile *Tor1a*$^{+/+}$ and *Tor1a*$^{+/\Delta gag}$ mice. The NMDAR antagonist MK-801 isolates the AMPAR-mediated EPSC component (black trace), while the NMDAR-EPSC (grey trace) is obtained by digital subtraction of the AMPAR EPSC from the dual-component EPSC (red). (*Right*) Summary plot of NMDA/AMPA current ratio calculated in SPNs from P26 *Tor1a*$^{+/+}$ and *Tor1a*$^{+/\Delta gag}$ mice. A significant decrease of NMDA/AMPA ratio was detected in P26 *Tor1a*$^{+/\Delta gag}$ mice, compared to *Tor1a*$^{+/+}$ (*Tor1a*$^{+/+}$, 2.92 ± 0.38, N = 3, n = 8; *Tor1a*$^{+/\Delta gag}$, 1.81 ± 0.25, N = 3, n = 6; Student's t test, p<0.05). (B) (*Left*) Representative EPSCs traces recorded at HP =+40 mV from SPNs of adult *Tor1a*$^{+/+}$ and *Tor1a*$^{+/\Delta gag}$ mice. (*Right*) Summary plot of NMDA/AMPA current ratio showing no significant difference between genotypes (*Tor1a*$^{+/+}$, 1.75 ± 0.15, N = 3, n = 7; *Tor1a*$^{+/\Delta gag}$, 2.01 ± 0.12, N = 3, n = 7; Student's t test, p>0.05). (C) AMPAR-mediated currents recorded at different HP in P26 *Tor1a*$^{+/+}$ and *Tor1a*$^{+/\Delta gag}$ SPNs. The IV relationship shows a significant increase in the current recorded at more hyperpolarized range from P26 *Tor1a*$^{+/\Delta gag}$ SPNs (HP=−70 mV: two-way ANOVA, *p<0.01). (*Left*) Summary plot of rectification index values of P26 *Tor1a*$^{+/+}$ and *Tor1a*$^{+/\Delta gag}$ SPNs (*Tor1a*$^{+/+}$, 0.50 ± 0.07, n = 7; *Tor1a*$^{+/\Delta gag}$, 0.43 ± 0.04, n = 8; Student's t test p>0.05). (D) AMPAR-mediated currents recorded in the presence of the GluA2-lacking AMPAR antagonist NASPM at P26. HP =−70 mV; to-way ANOVA, *p<0.01). (*Left*) Summary plots of the rectification index measured at P26 (*Tor1a*$^{+/+}$, 0.53 ± 0.04, n = 5, N = 6; *Tor1a*$^{+/\Delta gag}$, 0.46 ± 0.03, n = 7; Student's t test, p>0.05). (E) Normalized IV relationships of NMDAR-mediated currents show no difference between genotypes at P26 (two-way ANOVA, p>0.05). (F) Representative NMDA-mediated EPSCs recorded at HP = +40 mV from P26 SPNs. (G) Summary plots display rise and decay time of NMDA-EPSCs recorded at HP =+40 mV in SPNs from P26 *Tor1a*$^{+/+}$ and *Tor1a*$^{+/\Delta gag}$ mice (rise time: *Tor1a*$^{+/+}$, 7.78 ± 0.42, n = 9; *Tor1a*$^{+/\Delta gag}$, 9.23 ± 1.37, n = 7; Student's t test p>0.05; decay time: *Tor1a*$^{+/+}$, 502.50 ± 20.06, n = 9; *Tor1a*$^{+/\Delta gag}$, 422.10 ± 30.15, n = 7, Student's t test, *p<0.05). Values are presented as mean ± SEM.

*Figure 3 continued on next page*

*Figure 3 continued*

DOI: https://doi.org/10.7554/eLife.33331.006

The following source data is available for figure 3:

**Source data 1.** Electrophysiological characterization of AMPAR and NMDAR currents at corticostriatal synapses of SPNs in both $Tor1a^{+/+}$ and $Tor1a^{+/\Delta gag}$ mice.

DOI: https://doi.org/10.7554/eLife.33331.007

(100 μM). No significant difference in the RI of P26 SPNs was measured in the presence of NASPM (*Figure 3D*; p>0.05); yet, at hyperpolarized voltage ranges AMPAR-mediated current was still increased in $Tor1a^{+/\Delta gag}$ SPNs (*Figure 3D*; two-way ANOVA, p<0.01 at HP =−70 mV). These results further excluded possible alterations of AMPAR surface composition.

The normalized IV relationship of NMDAR-EPSCs showed the characteristic 'J-shape' (*Mayer et al., 1984*) in SPNs recorded at P26 from both genotypes (*Figure 3E*). No significant difference was found in the voltage-dependence of NMDARs (p>0.05). By analyzing the kinetics of the response at HP =+ 40 mV, we detected a significantly decreased decay time in $Tor1a^{+/\Delta gag}$ mice compared to controls (*Figure 3F,G*; p<0.05), despite a comparable rise time, suggesting a modification of NMDAR subunit composition (*Paoletti et al., 2013*). In particular, it is well-established that the decay time of NMDAR currents is correlated to the amount of GluN2-type subunits. GluN2A and GluN2B represent the most abundant NMDAR regulatory subunits expressed in SPNs (*Chen and Reiner, 1996*; *Dunah and Standaert, 2003*) and are characterized by a fast and slow decay time, respectively (*Sanz-Clemente et al., 2013*).

Taking into account all the above-described electrophysiological results, we evaluated the levels of AMPAR and NMDAR subunits into TIF fractions purified from striata of both juvenile (P26) and adult (P60) mice by means of WB analysis. We found a significant increase in the levels of both GluA1 and GluA2 AMPAR subunits in the postsynaptic compartment of P26 $Tor1a^{+/\Delta gag}$ mice compared to controls (*Figure 4A,B*; p<0.05), consistent with the observed reduction of the NMDA/AMPA ratio and the absence of any alteration of the RI (see *Figure 3*). Interestingly, we also found an increase of phosphorylation at GluA1-Ser845 (*Figure 4A,B*;, p<0.05), which is known to be correlated with LTP expression and to prevent endocytosis of GluA1-containing AMPARs (*Oh et al., 2006*; *Bassani et al., 2013*). Moreover, in agreement with the reduction of the NMDAR decay time, we observed an increase of GluN2A but not GluN2B subunit at postsynaptic sites of P26 $Tor1a^{+/\Delta gag}$ mice compared to $Tor1a^{+/+}$ (*Figure 4C,D*; p<0.05). Finally, no modifications of PSD-95, the most abundant scaffolding protein at the excitatory synapse, was observed (*Figure 4C,D*). Notably, these alterations of AMPAR and NMDAR subunits were not present in SPNs from P60 $Tor1a^{+/\Delta gag}$ mice (*Figure 4*; p>0.05).

Next, we performed a detailed evaluation of dendritic spine density and morphology in $Tor1a^{+/\Delta gag}$ SPNs, compared to age-matched $Tor1a^{+/+}$ mice. P26 $Tor1a^{+/\Delta gag}$ SPNs (*Figure 5A–D*) exhibited a higher number of mushroom-type spines (*Figure 5C*; p<0.05) and, consequently, a concomitant overall increase of dendritic spine width compared to $Tor1a^{+/+}$ mice (*Figure 5B*; p<0.05), thus suggesting an advanced stage of spine maturation, in agreement with the observed molecular GluN2A/GluN2B switch (see *Figure 4*). This event was associated, as expected, to an overall decrease of dendritic spine density (*Figure 5A*; p<0.05).

Conversely, P60 $Tor1a^{+/\Delta gag}$ mice (*Figure 5E–H*) showed a normalization of dendritic spine density (*Figure 5E*; p>0.05) and of spine width (*Figure 5F*; p>0.05) compared to $Tor1a^{+/+}$ mice. Furthermore, with respect to P26, at P60 the number of mushrooms remained unchanged in $Tor1a^{+/\Delta gag}$ mice but increased in $Tor1a^{+/+}$ (*Figure 5G*; p<0.05). Yet, at P60 $Tor1a^{+/\Delta gag}$ mice showed an increase of thin spines compared to $Tor1a^{+/+}$ mice (*Figure 5G*; p<0.05).

## Increased BDNF protein expression in Tor1a$^{+/\Delta gag}$ striatum at P26

Neurotrophic factors play a fundamental role in the development of SPNs and synaptic plasticity maturation (*Altar et al., 1997*; *Rauskolb et al., 2010*). Particularly, BDNF contributes to the developmental expression of AMPAR subunits at postsynaptic compartments (*Jourdi et al., 2003*; *Jourdi and Kabbaj, 2013*). The majority of BDNF, anterogradely transported to the striatum, originates from the cortex, where its expression begins in the first postnatal days (*Baydyuk and Xu,*

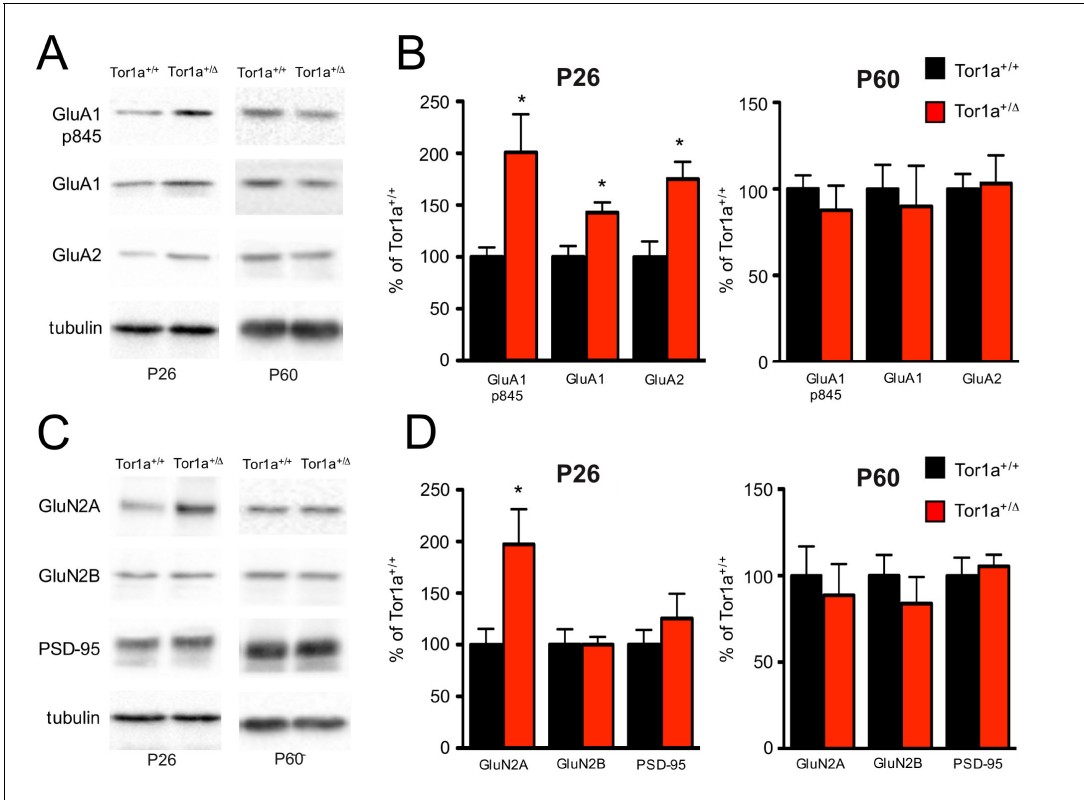

**Figure 4.** Molecular analysis of the SPNs postsynaptic compartment in P26 and P60 *Tor1a*$^{+/\Delta gag}$ compared to age-matched wild-type mice. WB analyses were performed on the post-synaptic TIF fraction in a minimum of three different animals per genotype. (**A**) WB analysis for GluN2A, GluN2B, PSD-95 and tubulin in P26 (left panel) and P60 (right panel) *Tor1a*$^{+/\Delta gag}$ and age-matched *Tor1a*$^{+/+}$ mice. (**C**) WB analysis for GluA1, GluA1p845, GluA2 and tubulin in P26 (left panel) and P60 (right panel) *Tor1a*$^{+/\Delta gag}$ and age-matched *Tor1a*$^{+/+}$ mice. (**B,D**) The histogram shows the quantification of protein levels following normalization on tubulin (P26 *Tor1a*$^{+/\Delta gag}$ compared to *Tor1a*$^{+/+}$, GluA1: 142.8 ± 9.8%, n = 5, p<0.05; GluA1-p845: 200.9 ± 36.6%, n = 5, p<0.05; GluA2: 175.1 ± 16.6%, n = 5, p<0.05; GluN2A: 197.3 ± 34.0%, n = 5, p<0.05; P60 *Tor1a*$^{+/\Delta gag}$ GluA1: 90.0 ± 23.4%, n = 5, p>0.05; GluA1-p845: 77.7 ± 14.2%, n = 5, p>0.05; GluA2: 103.2 ± 16.2%, n = 5, p>0.05; GluN2A: 88.8 ± 18.0%, n = 5,p>0.05). All values are mean ± SEM expressed as % of *Tor1a*$^{+/+}$ mice.

DOI: https://doi.org/10.7554/eLife.33331.008

*2014*). We first performed a WB time-course analysis of BDNF protein level in P15, P26 and adult (P60-P75) striatum. BDNF expression profile showed a similar age-dependent time-course in both genotypes (*Figure 6A,B*; P15 *vs* P26: *Tor1a*$^{+/+}$ p<0.05; *Tor1a*$^{+/\Delta gag}$p<0.01). As indicated by the BDNF/proBDNF ratio, in line with previous evidence (*Zermeño et al., 2009*), BDNF was highly expressed at P15 in both strains. At P26 the signal decreased, and then reached intermediate values in adults (*Figure 6A,B*).

Next, we compared striatal proBDNF and BDNF protein levels between genotypes at P26. In line with previous evidence, proBDNF was detected as a double band at ~32 KDa, whereas mature BDNF as a single band at 14 KDa (*Hartog et al., 2009*; *Koshimizu et al., 2009*; *Mandel et al., 2009*; *Tropea et al., 2011*) (*Figure 6C*). Both proBDNF and BDNF levels were increased at P26 in *Tor1a*$^{+/\Delta gag}$ striatum (*Figure 6C*; proBDNF p<0.01, BDNF p<0.05). We therefore examined Bdnf mRNA expression in *Tor1a*$^{+/\Delta gag}$ cortex. Quantitative PCR revealed an increased Bdnf expression in *Tor1a*$^{+/\Delta gag}$ cortex as compared to *Tor1a*$^{+/+}$ (*Figure 6D*; p<0.05). No significant difference between genotypes was measured in the proBDNF and BDNF striatal protein levels in adult mice (*Figure 6E*; p>0.05). Collectively, these data indicate an increase of BDNF level in P26 *Tor1a*$^{+/\Delta gag}$ striatum.

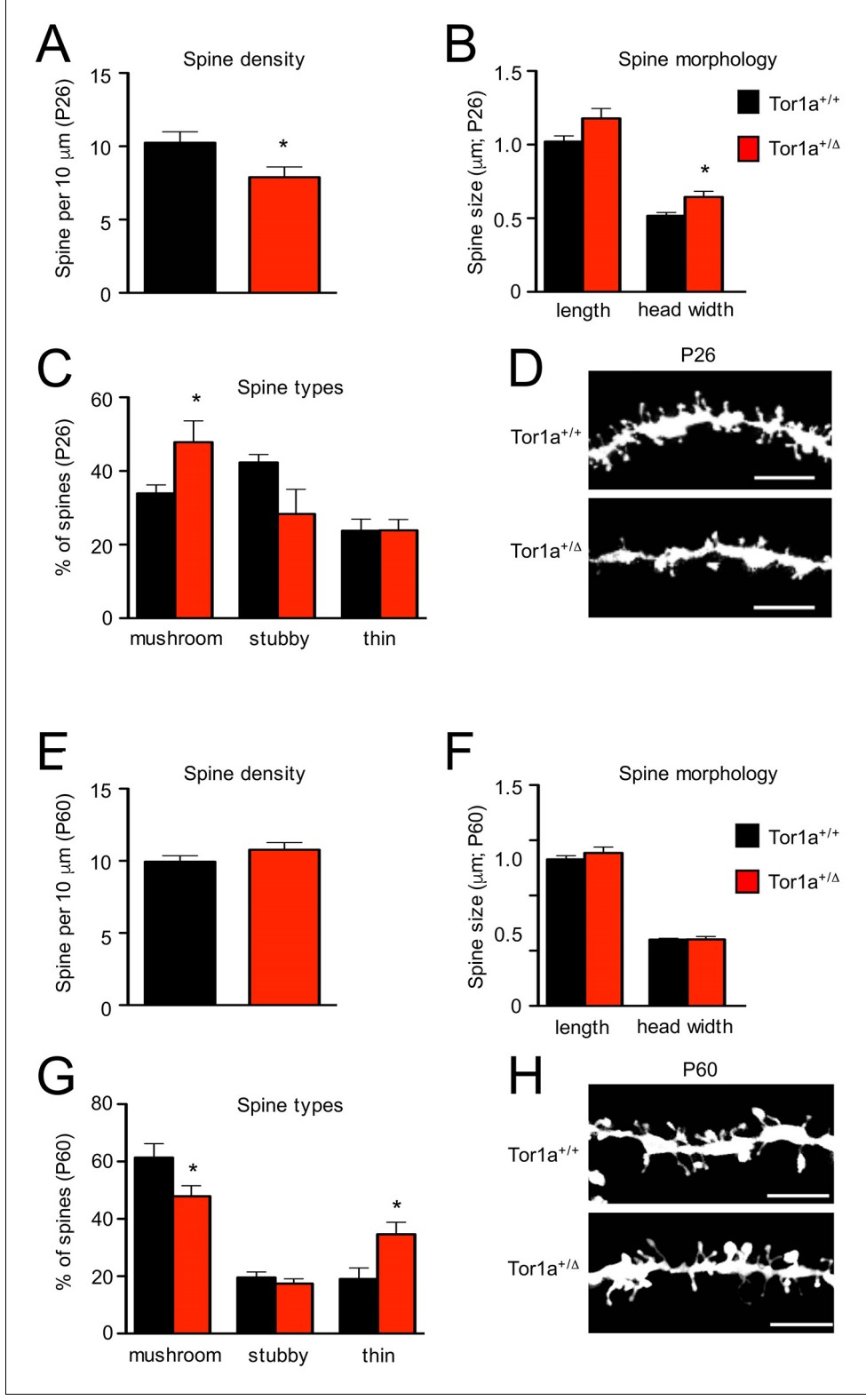

**Figure 5.** Analysis of dendritic spines morphology in P26 and P60 *Tor1a*$^{+/\Delta gag}$ compared to age-matched *Tor1a*$^{+/+}$ mice. (**A**) Histogram representing dendritic spine density in P26 *Tor1a*$^{+/\Delta gag}$ and *Tor1a*$^{+/+}$ mice (*Tor1a*$^{+/+}$, 10.25 ± 0.75 spines/10 μm, n = 10; *Tor1a*$^{+/\Delta gag}$, 7.89 ± 0.70 spines/10 μm, n = 10; unpaired Student's t test *p<0.05). (**B,C**) Histograms showing the quantification of dendritic spine size (B, spine length and head width) and dendritic spine type (C, mushroom, stubby, thin) in P26 *Tor1a*$^{+/\Delta gag}$ compared to *Tor1a*$^{+/+}$ mice (dendritic spine

*Figure 5 continued on next page*

*Figure 5 continued*

width $Tor1a^{+/+}$, 0.51 ± 0.02 μm, n = 10; $Tor1a^{+/\Delta gag}$, 0.64 ± 0.04 μm, n = 10, unpaired Student's t-test *p<0.05; mushroom-type spines $Tor1a^{+/+}$, 33.92 ± 2.32%, n = 10; $Tor1a^{+/\Delta gag}$, 47.81 ± 5.79%, n = 10, unpaired Student's t-test *p<0.05). (D) Representative images show dendrites of P26 $Tor1a^{+/\Delta gag}$ and $Tor1a^{+/+}$ mice. (E) Histogram representing dendritic spine density in P60 $Tor1a^{+/\Delta gag}$ and $Tor1a^{+/+}$ mice ($Tor1a^{+/+}$, 9.94 ± 0.41 spines/10 μm, n = 10; $Tor1a^{+/\Delta gag}$, 10.76 ± 0.50 spines/10 μm, n = 10; unpaired Student's t-test p>0.05). (F,G) Histograms showing the quantification of dendritic spine size (F, spine length and head width) and dendritic spine type (G, mushroom, stubby, thin) in P60 $Tor1a^{+/\Delta gag}$, compared to $Tor1a^{+/+}$ mice (spine width $Tor1a^{+/+}$, 0.600 ± 0.012 μm, n = 10; $Tor1a^{+/\Delta gag}$, 0.602 ± 0.027 μm, n = 10; p>0.05; mushroom-type spines $Tor1a^{+/+}$, 61.40 ± 4.81%, n = 10; $Tor1a^{+/\Delta gag}$, 47.92 ± 3.67%, n = 10; *p<0.05; thin spines $Tor1a^{+/+}$, 19.04 ± 3.85%, n = 10; $Tor1a^{+/\Delta gag}$, 34.64 ± 4.16%, n = 10; *p<0.05; unpaired Student's t-test). (H) Representative images show dendrites of P60 $Tor1a^{+/\Delta gag}$ and $Tor1a^{+/+}$ mice. Data were collected in a minimum of three different animals per genotype.
DOI: https://doi.org/10.7554/eLife.33331.009

The following source data is available for figure 5:

**Source data 1.** Analysis of dendritic spines morphology in P26 $Tor1a^{+/\Delta gag}$ compared to age-matched $Tor1a^{+/+}$ mice.
DOI: https://doi.org/10.7554/eLife.33331.010

**Source data 2.** Analysis of dendritic spines morphology in P60 $Tor1a^{+/\Delta gag}$ compared to age-matched $Tor1a^{+/+}$ mice.
DOI: https://doi.org/10.7554/eLife.33331.011

## BDNF regulates surface AMPA receptor expression and synaptic plasticity

BDNF has been shown to contribute to LTP induction in normal mice (*Jia et al., 2010*). To test whether the increase in BDNF levels was involved in the abnormal regulation of AMPA currents and in the synaptic plasticity deficits, BDNF signalling was selectively blocked by the tropomyosin-related kinase B (TrkB) receptor competitive antagonist, ANA-12 (*Cazorla et al., 2011*). A single in vivo administration of ANA-12 (0.5 mg/kg, intraperitoneal, 4 hr before the experiment) failed to rescue synaptic plasticity deficits in young $Tor1a^{+/\Delta gag}$ mice (data not shown). However, repetitive treatment with ANA-12 (0.5 mg/kg, intraperitoneal, 12 hr and 4 hr before the experiment; *Cazorla et al., 2011*; *Stragier et al., 2015*), completely rescued corticostriatal LTD expression in P28-35 $Tor1a^{+/\Delta gag}$ mice (*Figure 7A*; p<0.05). Additionally, ANA-12 treatment reduced LTP amplitude in P24-35 $Tor1a^{+/\Delta gag}$ mice (*Figure 7B*; p>0.05). In vehicle-treated $Tor1a^{+/+}$ and $Tor1a^{+/\Delta gag}$ mice, no significant change was observed (data not shown).

Likewise, in vivo treatment with ANA-12 totally normalized the IV curve of AMPAR-EPSC in P26 $Tor1a^{+/\Delta gag}$ mice (*Figure 7C*; p>0.05). Accordingly, also the RI displayed no significant difference between genotypes (*Figure 7C*; p>0.05). These findings suggest that increased BDNF levels are involved in the abnormal developmental expression of AMPARs on SPN postsynaptic membranes, leading to synaptic plasticity alterations in juvenile mice.

Finally, to demonstrate that BDNF alterations occur in a defined time-window, we tested the effect of ANA-12 on corticostriatal LTD expression in adult $Tor1a^{+/\Delta gag}$ mice. In vivo treatment with ANA-12 (0.5 mg/kg, intraperitoneal, two administrations at 12 hr and 4 hr before the experiment) failed to restore corticostriatal LTD in P60 $Tor1a^{+/\Delta gag}$ mice (*Figure 7D*; p>0.05) confirming that BDNF-dependent alterations are limited to a sensitive period.

## Cholinergic transmission is not involved in the early phase of synaptic plasticity alterations

Previous work demonstrated a prominent involvement of cholinergic transmission in the impairment of striatal synaptic plasticity in adult $Tor1a^{+/\Delta gag}$ mice (*Maltese et al., 2014*). To verify whether plasticity alterations in juvenile $Tor1a^{+/\Delta gag}$ mice could also involve cholinergic signaling, slices were pre-treated with the M1 muscarinic receptor antagonist, pirenzepine (100 nM, 20 min). Pirenzepine failed to rescue the expression of LTD in $Tor1a^{+/\Delta gag}$ mice from p28 to p35 (*Figure 7E*; p>0.05), indicating that distinct mechanisms underlie plasticity alterations at different developmental stages.

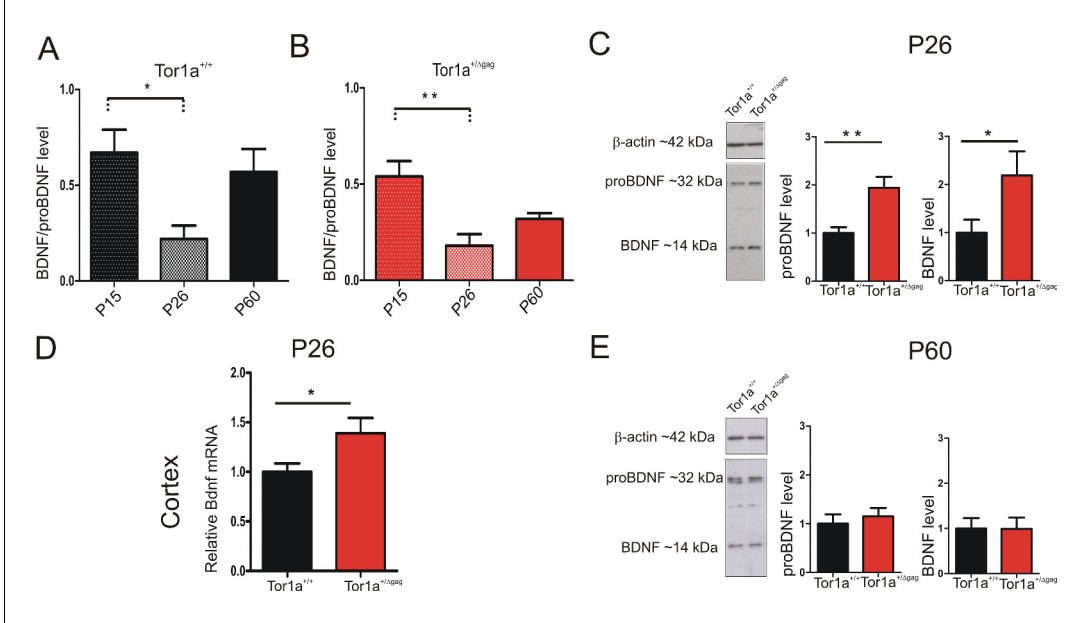

**Figure 6.** BDNF protein expression in the striatum of *Tor1a*[+/+] and *Tor1a*[+/Δgag] mice. (**A, B**) Striatal BDNF protein expression in *Tor1a*[+/+] and *Tor1a*[+/Δgag] mice at postnatal stages (**P15, P26, P60**). The graphs show the quantification of BDNF/proBDNF ratio at the various ages. Data are represented as mean ± SEM (*Tor1a*[+/+] P15: 0.67 ± 0.12, N = 4; P26: 0.22 ± 0.08, N = 4; P60: 0.57 ± 0.14, N = 3; *Tor1a*[+/Δgag] P15: 0.54 ± 0.08, N = 4; P26: 0.18 ± 0.06, N = 4; P60: 0.32 ± 0.03, N = 4; one-way ANOVA, *p<0.05; **p<0.01). (**C**) (*Left*) Representative WB of proBDNF and BDNF protein levels relative to β-actin in striatal extracts (30 μg) derived from P26 *Tor1a*[+/+] and *Tor1a*[+/Δgag] mice. (*Right*) The graphs show the quantitative analysis. The amount of proBDNF and BDNF was quantified relative to β-actin and normalized to wild-type mice. Data are represented as mean ± SEM (proBDNF *Tor1a*[+/+] 1.00 ± 0.12, n = 10; Tor1a[+/Δgag]1.95 ± 0.29, n = 8; BDNF Tor1a[+/+]: 1.00 ± 0.28, n = 8, *Tor1a*[+/Δgag]2.19 ± 0.50, n = 8, Student's t test: *p<0.05; **p<0.01). (**D**) Bdnf mRNA is upregulated in the cortex of *Tor1a*[+/Δgag] determined by qRT-PCR. The 2[-ΔΔCt] method was used to determine the relative expression, and all of the values are expressed relative to the levels of the wild-type mice as mean ± SEM (*Tor1a*[+/+] 1.000 ± 0.084, n = 10; *Tor1a*[+/Δgag]1.399 ± 0.163, n = 8; Student's t test: *p<0.05). (**E**) (*Left*) Representative Western blots of proBDNF and BDNF proteins relative to β-actin in striatal extracts (15 μg) derived from *Tor1a*[+/+] and *Tor1a*[+/Δgag] adult mice. (*Right*) The graphs show the quantitative analysis. The amount of proBDNF and BDNF was quantified relative to β-actin and normalized to wild-type mice. Data are represented as mean ± SEM (proBDNF *Tor1a*[+/+] 1.00 ± 0.19, n = 7, *Tor1a*[+/Δgag]1.15 ± 0.17, n = 7, p>0.05; BDNF *Tor1a*[+/+]: 1.00 ± 0.23 n = 7, *Tor1a*[+/Δgag]0.99 ± 0.25, n = 7, Student's t test: p>0.05).

DOI: https://doi.org/10.7554/eLife.33331.012

The following source data is available for figure 6:

**Source data 1.** BDNF protein expression in the striatum of *Tor1a*[+/+] and *Tor1a*[+/Δgag] mice.

DOI: https://doi.org/10.7554/eLife.33331.013

# Discussion

The critical period for the onset of symptoms in DYT1 dystonia patients matches an early time-window of activity-dependent plastic changes in the striatum, which shape motor memory and learning processes during childhood and early adolescence.

Our systematic analysis of functional and structural synaptic plasticity in DYT1 dystonia demonstrates: (i) The existence of a critical period when SPNs exhibit premature LTP; (ii) A significant increase of AMPAR levels in the postsynaptic compartment which correlates with the reduction of NMDA/AMPA ratio, the increased amplitudes of postsynaptic currents and the rightward shift in the AMPA I-V curve observed in juvenile *Tor1a*[+/Δgag] mice; (iii) A BDNF time-dependent increase in expression profile, which parallels the alterations described; (iv) abnormal plasticity is associated with profound changes of dendritic spine density and morphology in juvenile *Tor1a*[+/Δgag]; (v) A rescue of the synaptic plasticity deficits is obtained by in vivo administration of a TrkB inhibitor.

It is currently unknown why penetrance is only 30% in DYT1 mutation carriers. One possibility is that at circuit level, motor system is already impaired early during development. The existence of a defined period in which neurons are particularly susceptible to experience-driven modifications is well-established, concurrently with structural modifications, and is currently believed to represent a

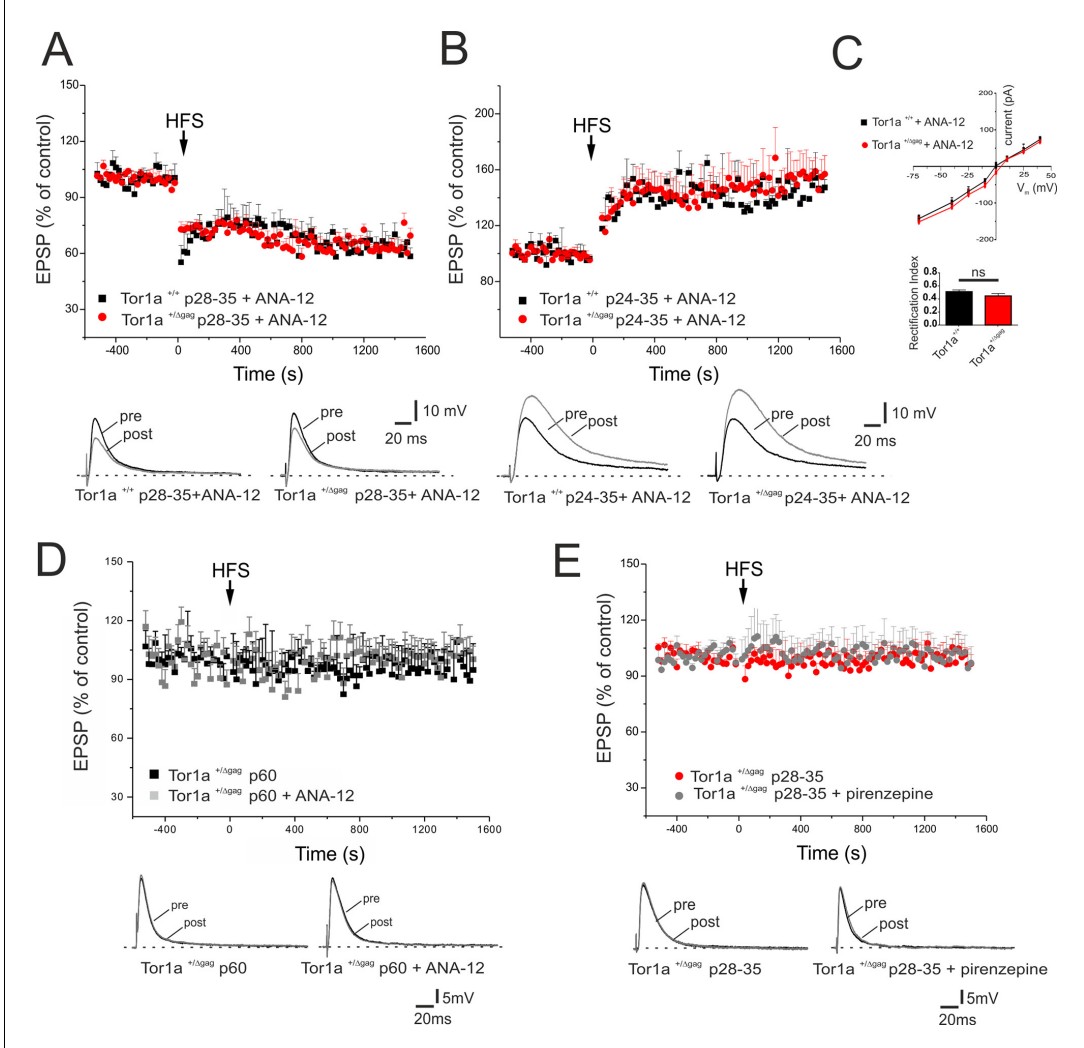

**Figure 7.** In vivo ANA-12 treatment rescues synaptic plasticity deficits in juvenile *Tor1a*[+/Δgag] mice. (**A**) Time-course of corticostriatal LTD in juvenile *Tor1a*[+/+] and *Tor1a*[+/Δgag] mice (P28-35): after in vivo treatment with the TrkB antagonist ANA-12, the HFS protocol (arrow) induces corticostriatal LTD expression in juvenile *Tor1a*[+/Δgag] mice (*Tor1a*[+/+] P28-35, 65.31 ± 1.44% of control; N = 3, n = 12, p<0.05; *Tor1a*[+/Δgag] P28-35, 63.41 ± 4.39% of control; N = 3, n = 10; paired Student's t test p<0.05). (*Bottom*) Representative EPSPs recorded before (pre) and 20 min after (post) HFS protocol. (**B**) Time-course of corticostriatal LTP after in vivo ANA-12 treatment: LTP displays a physiological amplitude in SPNs from in P24-35 *Tor1a*[+/Δgag] compared to wild-type littermates (*Tor1a*[+/+] P24-35, 144.55 ± 2.67% of control; N = 3, n = 8; *Tor1a*[+/Δgag] P24-35, 148.11 ± 10.55% of control; N = 3, n = 9; *Tor1a*[+/Δgag] *vs. Tor1a*[+/+] Student's t test p>0.05). (*Bottom*) Sample traces recorded pre and post LTP induction. (**C**) AMPAR-mediated currents recorded from P26 SPNs at HP from −70 mV to + 40 mV after in vivo treatment of *Tor1a*[+/+] and *Tor1a*[+/Δgag] mice with ANA-12. The treatment normalizes the current-voltage relationship in *Tor1a*[+/Δgag] neurons (HP=−70 mV: 2-way ANOVA p>0.05) and the rectification index (*Tor1a*[+/+], 0.51 ± 0.03, N = 3, n = 3; *Tor1a*[+/Δgag], 0.45 ± 0.04, N = 3, n = 5; Student's t test p>0.05) (**D**) In vivo treatment with ANA-12 does not restore corticostriatal LTD in adult (P60) SPNs recorded from *Tor1a*[+/Δgag] mice (vehicle: 95.66 ± 9.09% of control, N = 3, n = 8; ANA-12: 98.75 ± 11% of control, N = 3, n = 4; paired Student's t test p>0.05). (**E**) Slice pre-treatment with pirenzepine (100 nM) does not rescue LTD expression in P28-35 *Tor1a*[+/Δgag] SPNs (vehicle: 101.54 ± 1.07% of control, N = 3, n = 3; pirenzepine: 100.34 ± 8.96% of control; N = 3, n = 3; paired Student's t test p>0.05). (*Bottom*) Superimposed traces of EPSPs recorded pre and 20 min post HFS delivery.

DOI: https://doi.org/10.7554/eLife.33331.014

The following source data is available for figure 7:

**Source data 1.** In vivo ANA-12 treatment rescues synaptic plasticity deficits in juvenile *Tor1a*[+/Δgag] mice.

DOI: https://doi.org/10.7554/eLife.33331.015

nodal mechanism both in physiological and pathological conditions (*Turrigiano and Nelson, 2004*; *Johnston, 2004*; *Meredith, 2015*; *Calabresi et al., 2016*).

Our electrophysiological assessment of synaptic plasticity identifies a rather narrow time-window, between P15 and P26 when striatal SPNs exhibit a premature LTP, whereas LTD cannot be evoked. Although it has to be cautiously reminded that these alterations match those described in adult DYT1 striata (*Martella et al., 2014*), the time-course performed in the present study indicates their abnormal early appearance, in a developmental phase when normal striatal SPNs do not yet exhibit long-lasting synaptic changes. Of interest, the loss of LTD was observed in a similar time frame in a novel model with a rare missense variant in the *Tor1a* gene (*Bhagat et al., 2016*).

Moreover, we describe, along with electrophysiological deficits, molecular and structural changes at striatal synapses that appear to be limited to a specific time-window. Striatal LTP either in mature tissue preparation or in the developing striatum is dependent on the activation of NMDAR (*Calabresi et al., 1992b*; *Partridge et al., 2000*), whereas LTD depends on AMPAR (*Calabresi et al., 1992a*). Our electrophysiological and biochemical characterization demonstrates an increase in currents mediated by AMPAR, consistent with the increased amplitude of mEPSCs, and additionally, the NMDAR/AMPAR ratio was significantly reduced in SPNs from DYT1 mice. A major mechanism regulating synaptic strength involves the balance between synaptic insertion and removal of glutamate receptors into the postsynaptic membrane (*Gong and De Camilli, 2008*). Specifically, loss of homeostatic regulation of excitatory synapses in distinct neuronal subtypes involve postsynaptic changes in accumulation of AMPAR (*Lissin et al., 1998*; *O'Brien et al., 1998*). Accordingly, we observed a significant increase of both GluA1 and GluA2 subunits of AMPARs in the post-synaptic compartment of P26 $Tor1a^{+/\Delta gag}$ mice compared to controls, suggestive of an increased surface expression of AMPARs. Of interest, the significant increase of the phosphorylation of GluA1-Ser845, a well-established correlate of LTP (*Oh et al., 2006*; *Bassani et al., 2013*) is consistent with the abnormal LTP expression measured in DYT1 mice. Moreover, GluA1-Ser845 (*Roche et al., 1996*) plays a key role in the synaptic delivery of GluA1-containing AMPARs by LTP (*Esteban et al., 2003*; *Bassani et al., 2013*) and is involved in surface reinsertion/stabilization of AMPARs (*Ehlers, 2000*), thus providing a molecular mechanism for the observed increase of AMPARs at postsynaptic membranes in P26 $Tor1a^{+/\Delta gag}$ mice compared to controls. Thus, we hypothesize that the loss of LTD may be related to the aberrant composition of striatal AMPARs observed in mutant mice.

The identification of increased AMPAR subunit levels in the postsynaptic compartment offers new opportunities to identify potential regulators of AMPAR turnover. Neurotrophins have been implicated in glutamatergic synapse development and plasticity, suggesting a potential role in postsynaptic proteins distribution (*Causing et al., 1997*; *McAllister et al., 1997*; *Kong et al., 2001*). Previous work elucidated the role of BDNF in the regulation of AMPAR expression and function, including synaptic AMPAR subunit trafficking (*Narisawa-Saito et al., 1999*; *Jourdi and Kabbaj, 2013*). Indeed, BDNF treatment acutely controls both AMPAR subunits and their scaffolding proteins trafficking, thereby modifying the strength of synaptic activity (*Minichiello et al., 1999*; *Mauceri et al., 2004*). Remarkably, we observed an enhancement of pro-BDNF and BDNF protein level in P26 $Tor1a^{+/\Delta gag}$ mice, which appears critical for the onset of abnormal neurophysiological phenotype in DYT1 dystonia. Consistently, we obtained a functional rescue of synaptic plasticity and AMPA-mediated currents with the competitive antagonist of BDNF TrkB receptor ANA-12 (*Cazorla et al., 2011*).

Activity-dependent synaptic plasticity as well as composition and activity of NMDARs and AMPARs strictly govern modifications of dendritic spine morphology, leading to a long-lasting structural plasticity. Yet, BDNF also plays a major role in spine maturation in several brain regions, including the striatum (*Baquet et al., 2004*; *Rauskolb et al., 2010*). Thus, the abnormal increase in BDNF expression fits with the abnormalities in spine morphology we observed. In P26 $Tor1a^{+/\Delta gag}$ mice, we measured an increase in mushroom spines, suggestive of a 'premature' maturation process accompanied by an overall decrease in the density of dendritic spines. It is well-known that expression patterns of the GluN2 subunits of NMDARs at dendritic spines change during the first postnatal weeks. In particular, GluN2A expression increases from the second postnatal week to become widely expressed and abundant throughout the brain (*Bellone and Nicoll, 2007*; *Gray et al., 2011*). Yet, in agreement with the reduction of the NMDAR decay time, we found an increase of postsynaptic GluN2A in P26 $Tor1a^{+/\Delta gag}$ mice suggesting a 'premature' GluN2A/GluN2B switch, thus indicating the existence of a molecular and morphological early maturation of the excitatory synapse in this

DYT1 model. Moreover, the existence of a close coordination between spine size and AMPAR levels at synaptic membranes has been previously reported (*Kopec et al., 2007*; *Malinverno et al., 2010*) and spine volume has been positively correlated with the strength of AMPAR-mediated synaptic transmission. Accordingly, in *Tor1a*$^{+/\Delta gag}$ mice we found a significant increase of spine head width, an increase in mushroom spines and a concomitant increase of both GluA1 and GluA2 subunits of AMPARs.

Most of the molecular and structural alterations described in juvenile DYT1 mice were not confirmed at our analyses performed in adult (P60) mice. Indeed, inhibition of BDNF with ANA-12 did not offset the plasticity deficits in adult mice. Additionally, the anticholinergic agent pirenzepine failed to rescue the plasticity deficits in juvenile animals, contrarily to what reported in adults (*Dang et al., 2012*; *Martella et al., 2014*), indicating that distinct mechanisms sustain the abnormal patterns of synaptic activity at different developmental ages. Future work is required to address the precise mechanisms governing this switch.

Collectively, we demonstrate that the rise of BDNF, in a restricted time-window, drives AMPA receptor composition changes and, consequently, structural modifications in spine morphology, resulting in the loss of homeostatic regulation of synaptic plasticity early in postnatal life.

Our hypothesis is also consistent with the clinical observation that the beneficial effects of Deep Brain Stimulation (DBS) in dystonic patients is more effective in young patients, as compared to patients implanted later in life (*Isaias et al., 2008*). Additionally, compared to the prompt efficacy observed in Parkinson's disease patients, weeks are commonly required to obtain symptomatic relief following DBS, and improvements may continue to be manifest over time (*Vercueil et al., 2001*; *Krauss, 2002*; *Vidailhet and Pollak, 2005*). It is plausible that severity of abnormal plasticity is related to disease duration, thus justifying the longer time required to erase aberrant plasticity patterns.

In a therapeutic perspective, these sensitive periods might be considered as temporal windows of opportunity, during which specific molecular steps could be targeted to prevent aberrant plasticity to develop.

# Materials and methods

**Key resources table**

| Reagent type (species) or resource | Designation | Source or reference | Identifiers | Additional information |
|---|---|---|---|---|
| Gene (*Mus musculus*) | Tor1a | MGI:1353568 | Gene ID: 30931 | official full name: torsin family 1, member A (torsin A) |
| Strain, strain background (*M. musculus*) | C57BL/6J mice | Charles River | catalog number B6JSIFE10SZ - C57BL/6J SPF/VAF; RRID:IMSR_JAX:000664 | |
| Genetic reagent (*M. musculus*) | heterozygous knock-in Tor1a$^{+/\Delta gag}$ | *Goodchild et al. (2005)* - | | maintained on the C57BL/6J background |
| Antibody | monoclonal anti-PSD-95 | Neuromab | clone (k28/43) - catalog number 75–028; RRID:AB_2292909 | dilution 1:2000 in I-Block |
| Antibody | monoclonal anti-GluN2B | Neuromab | clone 59/20 - catalog number 75–097; RRID:AB_10673405 | dilution 1:1000 in I-Block |
| Antibody | polyclonal anti-GluA1 | Merck Millipore | catalog number AB1504; RRID:AB_2113602 | dilution 1:1000 in I-Block |
| Antibody | polyclonal anti-phospho-GluA1 (Ser845) | Merck Millipore | catalog number 04–1073; RRID:AB_1977219 | dilution 1:1000 in I-Block |
| Antibody | polyclonal anti-GluN2A | Sigma-Aldrich | catalog number M264 RRID:AB_260485 | dilution 1:1000 in I-Block |
| Antibody | monoclonal anti-GluA2 | Neuromab | clone L21/32 - catalog number 75–002; RRID:AB_2232661 | dilution 1:1000 in I-Block |

*Continued on next page*

*Continued*

| Reagent type (species) or resource | Designation | Source or reference | Identifiers | Additional information |
|---|---|---|---|---|
| Antibody | monoclonal anti-α-tubulin | Sigma-Aldrich | clone DM1A - catalog number T9026; RRID:AB_477593 | dilution 1:5000 in I-Block |
| Antibody | goat anti-DARPP-32 | R and D system | catalog number AF6259; RRID:AB_10641854 | dilution 1:500 in I-Block |
| Antibody | mouse anti-Enkephalin | Millipore | catalog number MAB350; RRID:AB_2268028 | dilution 1:1000 in I-Block |
| Antibody | mouse anti-β-actin | Sigma Aldrich | catalog number A5441; RRID:AB_476744 | dilution 1:20000 in I-Block |
| Commercial assay or kit | Clarity Western ECL Substrate | BioRad | - | reagent used to visualize protein bands with Chemidoc Imaging System |
| Commercial assay or kit | ECL reagent | GEHealthcare | catalog number GERPN2232 | reagent used to visualize protein bands with membranes were exposed to film |
| Commercial assay or kit | TRI-reagent | Sigma Aldrich | catalog number T9424 | reagent used to RNA extraction |
| Commercial assay or kit | DNAase I | Invitrogen | catalog number AMPD1-1KT | reagent used for elimination of DNA from RNA |
| Commercial assay or kit | Transcriptor First Strand cDNA Synthesis Kit | Roche | catalog number 04379012001 | reagent used to reverse transcribe RNA |
| Commercial assay or kit | Extract-N-Amp™ Tissue PCR Kit | SIGMA | catalog number XNAT2 | genotyping primers UP- AGT CTG TGG CTG GCT CTC C; Low- CCT CAG GCTGCT CAC AAC C |
| Chemical compound, drug | ANA-12 | Sigma-Aldrich | catalog number SML0209 | in vivo administration |
| Chemical compound, drug | CNQX disodium salt | Tocris | catalog number 0190/10 | application in bath during electrophysiology analysis |
| Chemical compound, drug | (+)-MK 801 maleate | Tocris | catalog number 0924/10 | application in bath during electrophysiology analysis |
| Chemical compound, drug | Tetrodotoxin citrate (TTX) | Tocris | catalog number 1069/1 | application in bath during electrophysiology analysis |
| Chemical compound, drug | Picrotoxin | Tocris | catalog number 1128/1 | application in bath during electrophysiology analysis |
| Chemical compound, drug | Biocytin | Tocris | catalog number 3349/10 | electrodes filled with biocytin, versatile marker used for neuroanatomical investigations of neuron IHC |
| Chemical compound, drug | Naspm trihydrochloride | Tocris | catalog number 2766/10 | application in bath during electrophysiology analysis |
| Software, algorithm | ImageLab | BioRad | - | software used for quantification of protein bands in western blotting experiments |
| Software, algorithm | ImageJ software | NIH; *Schneider et al. (2012)* | RRID:SCR_003070 | software used for the quantification of protein bands in western blotting and confocal laser scanning microscope |
| Software, algorithm | ClampFit 9 | pClamp | Molecular Devices; RRID:SCR_011323 | data analysis |
| Software, algorithm | Origin 8.0 | Microcal | RRID:SCR_002815 | data analysis |
| Software, algorithm | Prism 5.3 | GraphPad | RRID:SCR_002798 | data analysis |

## Animal model

Studies were carried out in juvenile (P15-P35) and adult (P60-P75) knock-in $Tor1a^{+/\Delta gag}$ mice heterozygous for ΔE-torsinA, a mutation that removes a single glutamic acid residue (ΔE) from the torsinA protein, and in their wild-type $Tor1a^{+/+}$ littermates (*Goodchild et al., 2005*). Genotyping was performed as described (*Ponterio et al., 2018*). Animal breeding, on a C57Bl/6J background, and handling were performed in accordance with the guidelines for the use of animals in biomedical research provided by the European Union's directives and Italian laws (2010/63EU, D.lgs. 26/2014; 86/609/CEE, D.Lgs 116/1992). The experimental procedures were approved by Fondazione Santa Lucia and University Tor Vergata Animal Care and Use Committees, and the Italian Ministry of Health (authorization #223/2017-PR).

## Experimental design

Age- and sex-matched wild-type and mutant littermates were randomly allocated to experimental groups. Investigators performing experiments and data analysis were blind to knowledge of genotype and treatment. Each observation was obtained from an independent biological sample. For electrophysiology, each cell was recorded from a different brain slice. All data were obtained from at least two animals in independent experiments. Biological replicates are represented with 'N' for number of animals and 'n' for number of cells. Sample size for any measurement was based on the ARRIVE recommendations on refinement and reduction of animal use in research, as well as on our previous studies.

## Electrophysiology

### Brain slice preparation

Mice were sacrificed by cervical dislocation, brains removed and sliced with a vibratome (Leica Microsystems) in oxygenated Krebs' solution (in mM: 126 NaCl, 2.5 KCl, 1.3 MgCl2, 1.2 NaH2PO4, 2.4 CaCl2, 10 glucose, 18 NaHCO3). Coronal and parasagittal corticostriatal slices (200–300 μm) were incubated in Krebs' solution at room temperature for 30 min. Then, individual slices were transferred into recording chambers continuously superfused with Krebs' solution (32–33°C) saturated with 95% $O_2$ and 5% $CO_2$.

### Patch-clamp recordings

Recordings were performed with AxoPatch 200B amplifiers and pClamp 10.2 software (Molecular Devices). For voltage-clamp experiments, pipettes (2.5–5 MΩ) were filled with $Cs^+$ internal solution (in mM: 120 CsMeSO3, 15 CsCl, 8 NaCl, 10 TEA-Cl, 10 HEPES, 0.2 EGTA, 2 Mg-ATP, and 0.3 Na-GTP; pH 7.3 adjusted with CsOH; 300 mOsm). For whole-cell recordings of glutamatergic sEPSCs, SPNs were clamped at HP=−70 mV in the presence of the GABA$_A$ receptor antagonist PTX (50 μM). For GABAergic sIPSCs, SPNs were recorded at HP =+ 10 mV in MK801 (30 μM) and CNQX (10 μM) to block NMDARs and AMPARs, respectively. Both mEPSCs and mIPSCs were measured by adding 1 μM TTX. PPR was measured at HP=−70 mV in PTX by delivering two stimuli at 25–1000 ms ISI. Synaptic strength was analyzed by measuring the NMDAR/AMPAR ratio at HP =+ 40 mV in PTX. The AMPAR–mediated component of EPSC was isolated in MK-801 and the NMDAR component was obtained by digital subtraction of the AMPAR component from the dual-component EPSC (*Sciamanna et al., 2012*). The AMPAR and NMDAR IV relationships were measured in the presence of PTX *plus* MK-801 or CNQX, respectively. The RI was calculated as ratio of the mean EPSC amplitudes measured at +40 mV and −70 mV.

### Sharp-electrode recordings

Current-clamp recordings of SPNs were performed with intracellular electrodes filled with 2M KCl (30–60 MΩ). Corticostriatal EPSPs were recorded in PTX (50 μM). HFS (three trains 100 Hz, 3 s, 20 s apart) was delivered at suprathreshold intensity to induce LTD. Magnesium was omitted to optimize LTP induction (*Calabresi et al., 1992b*). The EPSP amplitude was averaged and plotted over-time as percentage of control pre-HFS amplitude.

## Gene expression analysis

P26 $Tor1a^{+/+}$ and $Tor1a^{+/\Delta gag}$ mouse cortex was collected in PCR clean tubes and stored at −80℃. Total RNA was isolated using TRI-reagent (Sigma Aldrich), quantified and treated with DNAase I (Invitrogen). Integrity was confirmed by 1% agarose gel electrophoresis. RNA was reverse-transcribed using random hexamer primer and anchored-oligo (dT)18 primer according to the manufacturer's instructions (Transcriptor First Strand cDNA Synthesis Kit, Roche). Real-time PCR was performed on 25 ng cDNA by using LightCycler 480 Probes Master (04707494001, Roche) with Roche Light Cycler LC480 system, Bdnf and Hprt1 primers designed on the Roche Universal Probe Library Assay Design Center: https://configurator.realtimeready.roche.com/assaysupply_cp/pages/singleAssays/searchResult.jsf

Raw Ct values for Bdnf gene were normalized to the endogenous control gene Hprt1. Technical triplicates were analyzed for all samples and mean values were utilized for statistical analysis. The relative expression was determined using the $2^{-\Delta\Delta Ct}$ method (*Livak and Schmittgen, 2001*).

## Immunohistochemistry

To identify direct- and indirect-pathway SPNs electrodes were loaded with biocytin, as described (*Martella et al., 2009*). Briefly, slices were fixed with 4% PFA in 0.12 M PB and 30 µm thick sections were cut from each slice with a freezing microtome, then dehydrated with serial alcohol dilutions to improve antigen retrieval and reduce background (*Buchwalow et al., 2011*). We used the following primary antibodies: goat anti-DARPP-32 (1:500 AF6259, R and D system), mouse anti-Enkephalin (1:1000 MAB350, Millipore), and secondary antibodies: anti-goat alexa 647 (Invitrogen), anti-mouse cyanine 3 (Jackson ImmunoResearch) and streptavidin-conjugated alexa 488 (Life Technologies). All sections used for analysis were processed together. Images were acquired with a LSM700 Zeiss confocal laser scanning microscope and analyzed with ImageJ software (NIH; *Schneider et al., 2012*). Noise was reduced by applying background subtraction in ImageJ.

## Subcellular fractionation and western blotting (WB)

To obtain a preparation that contains selectively proteins of the post-synaptic density (PSD), subcellular fractionation of striatal tissue was performed as reported (*Gardoni et al., 2006*; *Paillé et al., 2010*) with minor modifications. Briefly, striata were homogenized with a Teflon-glass potter in ice-cold 0.32M sucrose containing 1 mM HEPES pH 7.4, 1 mM $MgCl_2$, 1mM EDTA, 1 mM $NaHCO_3$, 0.1 mM phenylmethanesulfonylfluoride (PMSF) in the presence of a complete set of proteases and phosphatase inhibitors (Complete™ Protease Inhibitor Cocktail Tablets and PhosSTOP™ Phosphatase Inhibitor Cocktail, Roche Diagnostics). The homogenized tissue was centrifuged at 13,000 g for 15 min. The pellet was re-suspended in a buffer containing 75 mM KCl and 1% Triton X-100 and spun at 100,000 g for 1 hr. The final pellet, referred to as Triton-insoluble postsynaptic fraction (TIF), was homogenized in a glass-glass potter in 20 mM HEPES supplemented with Complete™ tablets and stored at −80℃ until use. Protein samples were separated onto an acrylamide/bisacrylamide gel at the appropriate concentration, transferred to a nitrocellulose membrane and immunoblotted with the appropriate primary and HRP-conjugated secondary antibodies. For WB analysis, the following unconjugated primary antibodies were used: polyclonal anti-GluN2A antibody (Sigma-Aldrich); monoclonal anti-GluN2B antibody (NeuroMab); polyclonal anti-GluA1 antibody (Merck Millipore); polyclonal anti-phospho-GluA1 (Ser845; Merck Millipore); monoclonal anti-GluA2 antibody (NeuroMab); monoclonal anti-PSD-95 antibody (NeuroMab); monoclonal anti-α-tubulin antibody (Sigma-Aldrich). Membrane development was performed with the reagent Clarity Western ECL Substrate (Bio-Rad) and labeling was visualized by Chemidoc Imaging System and ImageLab software (Bio-Rad). For quantification, each protein was normalized against the corresponding α-tubulin band.

WB analysis of BDNF on mouse striatum was performed as described (*Sciamanna et al., 2015*; *Ponterio et al., 2018*). Protein extracts (15–30 µg) were loaded with page LDS sample buffer (Invitrogen, Waltham, Massachusetts, USA) containing DTT and denatured at 95℃ for 5 min. Proteins were separated on 15% SDS-PAGE, and transferred onto 0.45 µm polyvinylidene fluoride (PVDF) membranes. The following primary antibodies were utilized: rabbit anti-BDNF (1:200 sc-546, SantaCruz Biotechnology) and mouse anti-β-actin (1:20.000 A5441, Sigma Aldrich), as loading control, followed by anti-rabbit or anti-mouse horseradish peroxidase (HRP)-conjugated secondary antibodies. Immunodetection was performed by ECL reagent (GEHealthcare) and membranes were

exposed to film (Amersham). Quantification of the band intensity on scanned filters was achieved by ImageJ software.

## Spine morphology

Carbocyanine dye DiI (Invitrogen) was used to label neurons as previously described (*Kim et al., 2007*; *Stanic et al., 2015*). Images were taken using an inverted LSM510 confocal microscope (Zeiss). For morphological analysis, cells were chosen randomly for quantification from four to eight different coverslips; images were acquired using the same settings/exposure times, and at least 10 cells for each condition were analyzed. Morphological analysis was performed with ImageJ software to measure spine density and size. For each dendritic spine the length, the head and neck width were measured, which was used to classify spines into categories (thin, stubby and mushroom) (*Harris et al., 1992*).

## Statistical analysis

Data were analysed with ClampFit 9 (pClamp, Molecular Devices), Origin 8.0 (Microcal) and Prism 5.3 (GraphPad) softwares. All data were obtained from at least two independent experiments and are represented as mean ± SEM. Statistical significance was evaluated, as indicated in figure legends, using paired and unpaired Student's t test, and one-way ANOVA with post-hoc Tukey test and two-way ANOVA with Bonferroni posttest for group comparisons. Statistical tests were two-tailed, the confidence interval was 95%, and the alpha-level used to determine significance was set at $p < 0.05$.

## Acknowledgements

We wish to thank Dystonia Medical Research Foundation for funding and for their work helping to increase understanding of this disease. Authors are grateful to Dr Nicole Calakos for critical reading of the manuscript and for helpful comments. We wish to thank also Elisa Zianni and Massimo Tolu for their skillful technical support. This work was supported by a PRIN 2010–2011 grant of the Ministero dell'Istruzione, dell'Università e della Ricerca to AP and FG.

## Additional information

### Funding

| Funder | Grant reference number | Author |
| --- | --- | --- |
| Ministero dell'Istruzione, dell'Università e della Ricerca | PRIN 2010-2011 | Fabrizio Gardoni<br>Antonio Pisani |
| Dystonia Medical Research Foundation | 2017 | Antonio Pisani |

The funders had no role in study design, data collection and interpretation, or the decision to submit the work for publication.

### Author contributions

Marta Maltese, Conceptualization, Data curation, Formal analysis, Investigation, Writing—original draft, Writing—review and editing; Jennifer Stanic, Data curation, Formal analysis, Investigation, Methodology, Writing—original draft; Annalisa Tassone, Paola Imbriani, Data curation, Formal analysis, Investigation, Methodology; Giuseppe Sciamanna, Data curation, Supervision, Investigation, Writing—review and editing; Giulia Ponterio, Data curation, Investigation, Methodology; Valentina Vanni, Investigation, Methodology; Giuseppina Martella, Formal analysis, Investigation, Methodology; Paola Bonsi, Fabrizio Gardoni, Conceptualization, Data curation, Writing—original draft, Writing—review and editing; Nicola Biagio Mercuri, Conceptualization, Supervision, Writing—review and editing; Antonio Pisani, Conceptualization, Data curation, Formal analysis, Supervision, Methodology, Writing—original draft, Writing—review and editing

## Author ORCIDs

Giuseppina Martella (iD) http://orcid.org/0000-0002-8927-7107
Paola Imbriani (iD) https://orcid.org/0000-0003-3373-5073
Paola Bonsi (iD) http://orcid.org/0000-0001-5940-9028
Fabrizio Gardoni (iD) http://orcid.org/0000-0003-4598-5563
Antonio Pisani (iD) http://orcid.org/0000-0002-8432-594X

## Ethics

Animal experimentation: Animal breeding and handling were performed in accordance with the guidelines for the use of animals in biomedical research provided by the European Union's directives and Italian laws (2010/63EU, D.lgs. 26/2014; 406 86/609/CEE, D.Lgs 116/1992). The experimental procedures were approved by Fondazione Santa Lucia and University Tor Vergata Animal Care and Use Committees and the Italian Ministry of Health (authorization #223/2017-PR).

## Decision letter and Author response

Decision letter https://doi.org/10.7554/eLife.33331.018
Author response https://doi.org/10.7554/eLife.33331.019

# Additional files

Supplementary files

• Transparent reporting form
DOI: https://doi.org/10.7554/eLife.33331.016

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
