## [Decision Letter]

Thank you for submitting your article "Early structural and functional plasticity alterations in a susceptibility period in DYT1 dystonia striatum" for consideration by *eLife*. Your article has been reviewed by two peer reviewers, and the evaluation has been overseen by a Reviewing Editor and a Senior Editor. The following individual involved in review of your submission has agreed to reveal his identity: Gilberto Fisone (Reviewer #1).

The reviewers have discussed the reviews with one another and the Reviewing Editor has drafted this decision to help you prepare a revised submission.

Summary:

This manuscript describes changes in neural plasticity in developing striatal spiny neurons in a mouse model of DYT1 dystonia. The studies presented here are among the first to explore these early developmental mechanisms. The results are interesting and valuable, because DYT1 dystonia has been attributed to maladaptive plasticity during an early developmental time window and the results provide novel insights into mechanisms that may be responsible for abnormal plasticity in DYT1 dystonia. The authors link the mutation to a premature upregulation of BDNF signalling in striatum. The studies define also a critical time window time in which cellular alterations at the post synaptic glutamate receptor composition and changes in LTP activity take place. Finally they show that the BDNF mediated alteration in stratal plasticity can be rescued by systemic treatment with TrkB antagonists. The manuscript data are strongly supportive for the interpretations of the authors, while some major rewriting in introduction and discussion are needed.

Essential revisions:

1) While not essential, it was suggested to test whether the early treatment with the TrkB antagonist, ANA-12, which rescues LTD in *Tor1a^+/Δgag^* mice at P28-35 (Figure 8A-C), also results in maintenance of LTD during adulthood (e.g. P60). This will indicate that, in DYT, an intervention limited to a specific developmental window may have long-term positive repercussions even later in life.

2) The Introduction and Discussion have statements that refer very broadly to abnormalities of neural plasticity in multiple neuropsychiatric disorders, where the data are often very sparse and speculative. These over-generalized statements about other disorders should be replaced with more background material supporting the maladaptive plasticity hypothesis specifically for DYT1 dystonia. There are many papers on this, especially showing abnormalities of plasticity in humans with DYT1 dystonia. Of course we cannot compare directly the prior human and current mouse studies, but providing this background for DYT1 gives a stronger rationale for the current study. This change would give the manuscript more focus, and make the rationale for studying neural plasticity in this disorder stronger.

---

## [Author Response]

Essential revisions:1) While not essential, it was suggested to test whether the early treatment with the TrkB antagonist, ANA-12, which rescues LTD in Tor1a^+/Δ^gag mice at P28-35 (Figure 8A-C), also results in maintenance of LTD during adulthood (e.g. P60). This will indicate that, in DYT, an intervention limited to a specific developmental window may have long-term positive repercussions even later in life.

The suggestion that a pharmacological treatment in a specific time window might have beneficial repercussions later in life is certainly fascinating, and we appreciate this suggestion. However, we realized that breeding of an appropriate number of animals would take much longer than expected. Moreover, the specific design of the pharmacological treatment with trkB receptor antagonists requires caution, for a number of reasons, such as treatment duration, route of administration, the delicate developmental phase, etc. Future work is required to specifically address this question.

2) The Introduction and Discussion have statements that refer very broadly to abnormalities of neural plasticity in multiple neuropsychiatric disorders, where the data are often very sparse and speculative. These over-generalized statements about other disorders should be replaced with more background material supporting the maladaptive plasticity hypothesis specifically for DYT1 dystonia. There are many papers on this, especially showing abnormalities of plasticity in humans with DYT1 dystonia. Of course we cannot compare directly the prior human and current mouse studies, but providing this background for DYT1 gives a stronger rationale for the current study. This change would give the manuscript more focus, and make the rationale for studying neural plasticity in this disorder stronger.

We thank the Editor and the reviewers for these suggestions, which allowed us to narrow our focus on plasticity and dystonia. We entirely revised the Introduction, summarizing the multiple findings reporting impaired plasticity in humans, and then linking these findings to the evidence obtained in animal models. The revision now highlights the remarkable similarity of plasticity abnormalities found in humans and mice and gives the Introduction more focus. Along the same line, the Discussion has been modified.